# Normalisation of brain connectivity through compensatory behaviour, despite congenital hand absence

Avital Hahamy[1], Stamatios N Sotiropoulos[2], David Henderson Slater[3], Rafael Malach[1], Heidi Johansen-Berg[2], Tamar R Makin[2]*

[1]Department of Neurobiology, Weizmann Institute of Science, Rehovot, Israel; [2]FMRIB Centre, Nuffield Department of Clinical Neurosciences, University of Oxford, Oxford, United Kingdom; [3]Oxford Centre for Enablement, Nuffield Orthopaedic Centre, Oxford, United Kingdom

**Abstract** Previously we showed, using task-evoked fMRI, that compensatory intact hand usage after amputation facilitates remapping of limb representations in the cortical territory of the missing hand (*Makin et al., 2013a*). Here we show that compensatory arm usage in individuals born without a hand (one-handers) reflects functional connectivity of spontaneous brain activity in the cortical hand region. Compared with two-handed controls, one-handers showed reduced symmetry of hand region inter-hemispheric resting-state functional connectivity and corticospinal white matter microstructure. Nevertheless, those one-handers who more frequently use their residual (handless) arm for typically bimanual daily tasks also showed more symmetrical functional connectivity of the hand region, demonstrating that adaptive behaviour drives long-range brain organisation. We therefore suggest that compensatory arm usage maintains symmetrical sensorimotor functional connectivity in one-handers. Since variability in spontaneous functional connectivity in our study reflects ecological behaviour, we propose that inter-hemispheric symmetry, typically observed in resting sensorimotor networks, depends on coordinated motor behaviour in daily life.

*For correspondence: tamar. makin@ndcn.ox.ac.uk

**Competing interests:** The authors declare that no competing interests exist.

**Reviewing editor**: Ranulfo Romo, Universidad Nacional Autonoma de Mexico, Mexico

## Introduction

The relationship between deprivation-triggered brain plasticity and adaptive behaviour following visual loss has been a focus of much research and debate (*Merabet and Pascual-Leone, 2010*; *Baseler et al., 2011*; *Espinosa and Stryker, 2012*). In the sensorimotor system, a link between brain plasticity following amputation and compensatory behaviour has only recently been established (*Makin et al., 2013a*; *Philip and Frey, 2014*). In people who were born without a hand, or who lost it in childhood, research on related behavioural changes has focused on compensatory foot usage (which is uncommon relative to arm usage), with conflicting results (*Yu et al., 2006*; *Stoeckel et al., 2009*). We recently showed using task-evoked fMRI that the way individuals use their bodies to compensate for their disability when missing a hand (either at birth or later in life due to injury) is reflected in the way the cortical territory of the missing hand is recruited during limb movements (*Makin et al., 2013a*). Specifically, individuals born with one hand (i.e., congenital unilateral upper limb absence; hereafter termed: 'one handers') showed both higher usage of the handless arm (i.e., residual arm) in daily tasks and increased activation in the cortical territory of the missing hand when moving that arm. In contrast, amputees that lost a hand later in life tended to use their remaining hand (i.e., intact hand) more frequently and also showed greater activation in the territory of the missing hand when moving the intact hand. These results provide evidence that whichever body part is being over-used to compensate for the absence of a hand also gains increased representation in the cortical territory of the missing hand.

Changes in brain activity underlying task performance may also be captured indirectly by measuring spontaneous fluctuations in fMRI signal while participants are not performing a task, but are instead resting in the scanner (*Gilaie-Dotan et al., 2013*; *Harmelech and Malach, 2013*). During rest, the human brain is characterised by activation patterns that are temporally coherent and reproducible across known functional networks, and this coherence is termed 'functional connectivity' (*Biswal et al., 1995*; *Nir et al., 2006*; *Smith et al., 2009*; *Yeo et al., 2011*). To date, the relationship between altered resting-state functional connectivity and compensatory behaviour in one-handers is not known (*Yu et al., 2014*). Since it has been hypothesised that patterns of resting-state functional connectivity reflect the activity patterns evoked by every-day behaviours (*Harmelech and Malach, 2013*), we predicted that the way individuals behave to compensate for their disability would also be reflected in resting-state connectivity patterns. We focused on inter-hemispheric functional connectivity patterns, which are stronger between homologous regions along the sensorimotor homunculus, resulting in a symmetrical connectivity pattern. We aimed to test whether this symmetry is disrupted in one-handers who have a more asymmetrical pattern of limb use in daily life (i.e., increased usage of their intact hand vs their residual arm).

Here we identified atypical asymmetry in functional connectivity and in structural features (as measured by white matter microstructure) in a relatively large sample of one-handers (n = 14, see *Table 1* for demographic details) compared with matched two-handed controls (n = 23). Importantly, the degree of functional connectivity between the two homologous hand regions in the sensorimotor cortex reflected the degree to which one-handers utilised their residual arm during daily activities. These findings extend our previous reports by demonstrating that resting-state connectivity alterations reflect adaptive behaviours beyond the scope that could be identified using task-evoked brain activity. These findings also demonstrate the imprinting of daily behaviours in spontaneous brain activity, as previously postulated (*Harmelech and Malach, 2013*).

## Results

### Increased anatomical asymmetry in the corticospinal tracts of one-handers

The corticospinal tracts mediate a range of descending pathways from cortical motor areas to the spinal cord, as well as ascending tracts to somatosensory cortex. To study structural alterations in motor-related pathways of one-handers, asymmetry in white matter microstructure of the bilateral corticospinal tracts was investigated. Fractional anisotropy (FA), an index commonly derived from diffusion MRI, was used as a probe of tissue microstructure. This measurement is thought to reflect myelination and fibre density, but is also influenced by other micro-features (e.g., membrane permeability, tissue geometry, see 'Discussion') (*Zatorre et al., 2012*; *Jones et al., 2013*; *Sampaio-Baptista et al., 2013*). One-handers showed increased asymmetry in white matter microstructure (reflected by decreased white matter FA in the corticospinal tract contralateral to the missing hand relative to the ipsilateral tract) compared to controls (p = 0.003, randomisation test, *Figure 1A*). The effect peaked at the posterior limb of the internal capsule (peak z coordinate = 88, p = 0.02 corrected, *Figure 1B*), an area that contains sensory thalamic fibres, but was also distributed across extensive regions of the tract (*Figure 1C*). No significant group differences were found for control tracts that are not directly associated with the sensorimotor system (Inferior fronto-occipital fasciculus: p = 0.28, Inferior longitudinal fasciculus: p = 0.27, randomisation tests).

Since accumulating evidence links white matter microstructure with motor behaviour (*Scholz et al., 2009*; *Sampaio-Baptista et al., 2013*), we aimed to test if asymmetry of white matter microstructure would be related to compensatory arm usage, as previously assessed (*Makin et al., 2013a*). However, no significant relationship was found (r = −0.27, p = 0.17, see 'Materials and methods').

### Increased functional asymmetry in one-handers

In the typical brain, the sensorimotor system is characterised by high levels of inter-hemispheric functional connectivity between homologous brain areas, resulting in symmetrical connectivity patterns (*Biswal et al., 1995*; *Yeo et al., 2011*). To study functional changes in one-handers, voxel-wise alterations in resting-state functional connectivity to the missing hand's territory (or the non-dominant hand region in controls) were examined using a seed-based approach (see 'Materials and methods'). We defined the seed-region based on the mirror image of task-evoked activation while participants moved their

**Table 1.** Demographic details of individuals with congenital limb absence

| Participant | Age | Level of limb deficiency | Affected limb |
|---|---|---|---|
| A01 | 33 | below elbow | Right |
| A02 | 26 | below elbow | Left |
| A03 | 37 | below elbow | Left |
| A04 | 33 | wrist | Left |
| A05 | 27 | below elbow | Left |
| A06 | 56 | below elbow | Left |
| A07 | 51 | wrist | Left |
| A08 | 24 | below elbow | Right |
| A09 | 51 | below elbow | Right |
| A10 | 20 | below elbow | Left |
| A11 | 48 | above elbow | Left |
| A12 | 24 | below elbow | Left |
| A13 | 35 | below elbow | Right |
| A14 | 48 | below elbow | Left |

dominant/intact hand (controls/one-handers, respectively). This seed-region encompassed both the primary somatosensory and motor cortices (SI and M1, respectively, see purple region in *Figure 2B–C*). Connectivity was assessed in one-handers compared with controls across the entire brain. Reduced connectivity to the seed-region was identified in the medial aspect of the homologous (intact) hand region, in the anterior bank of the central sulcus, corresponding with M1 (*Figure 2B*). This finding reflects an aberration in the typically symmetrical resting-state sensorimotor network. Group comparison of inter-hemispheric connectivity in the intact hand region, defined based on brain activations during hand movements, confirmed reduced connectivity between the two sensorimotor hand regions (p = 0.002, randomisation test, middle panel of *Figure 2B*). Smaller Clusters showing reductions in connectivity with the seed-region were also identified in the posterior aspect of the dorsomedial prefrontal cortex contralateral to the missing hand and in the middle temporal gyrus bilaterally (*Table 2*).

To ensure that this effect was not caused by reduced signal-to-noise ratio in the missing hand territory of one-handers, the variance (amplitude) of signal fluctuations within this region was compared between one-handers and controls. No differences were found in the variance of both the mean regional time-course (p = 0.5, randomisation test), or the mean single-voxel time-courses (p = 0.5, randomisation test).

## Resting state connectivity patterns reflect bimanual usage

We have previously shown that the missing hand territory of one-handers is activated when one-handers move their residual arm. Given that the missing hand territory (used here as a seed-region) was shown to represent the residual arm, It is possible that variations in connectivity strength between the hand regions of one-handers reflect compensatory arm usage. To address this, a whole brain voxel-wise analysis was conducted to test for positive correlations between functional connectivity values (to the missing hand's territory, as described above) and the level of residual arm usage in everyday activities (*Makin et al., 2013a*). Significant correlation with behavioural scores was found in a single cluster within the intact cortical hand region (*Figure 2C*), specifically in the anterior bank of the central sulcus, corresponding with M1. This result indicates that one-handers who employ their residual arm for typically bimanual tasks also demonstrate higher levels of hand region inter-hemispheric connectivity.

Finally, to confirm that use-dependent changes in connectivity reflected functional coupling between the two hand regions, this relationship was assessed by extracting connectivity values from the intact hand region, as defined using task-evoked fMRI. A correlation analysis confirmed that the more one-handers utilised their residual arm in daily tasks, the higher the inter-hemispheric

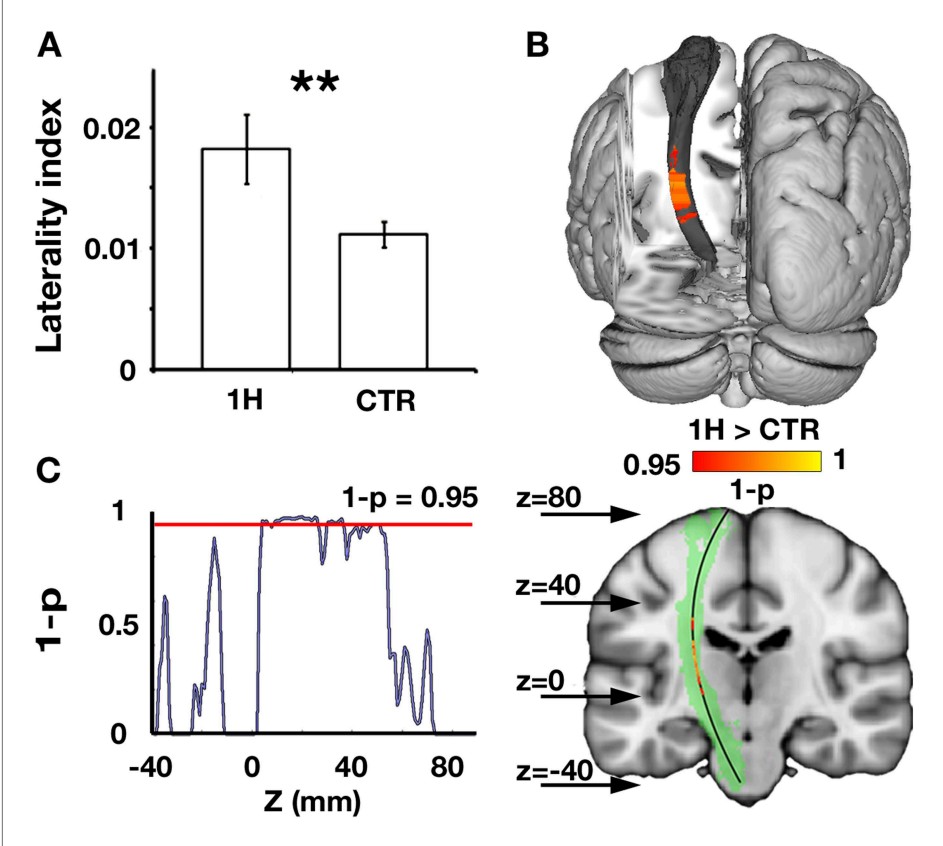

**Figure 1**. Increased structural asymmetry in one-handers. (**A**) White matter values (mean fractional anisotropy ± s.e.m.) were extracted from the bilateral corticospinal tracts of each participant, and laterality indices were computed to determine the level of white matter asymmetry across hemispheres ([(intact − residual)/(intact + residual)] for one-handers, and [(dominant − nondominant)/(dominant + nondominant)] for controls). Laterality indices were significantly higher in one-handers compared to controls. (**B**) To spatially identify regions of increased white-matter laterality, laterality indices were computed for each slice along the superior–inferior axis of the corticospinal tracts, and a group comparison was conducted using permutation-based cluster statistics. Regions showing significant group differences along the corticospinal tracts were centred around the posterior limb of the internal capsule, reflecting higher laterality in one-handers compared to controls. Results are presented on the left corticospinal tract from a posterior view. (**C**) Right: Regions that show significantly higher laterality values in one-handers are presented on the left corticospinal tract (highlighted in green) along with their spatial locations along the z axis, as marked by arrows. Left: The lateralisation profile of white matter microstructure along the corticospinal tract is represented by p-values, derived from the comparison between one-handers and controls. Higher values indicate stronger statistical differences. The red line marks the significance threshold, corrected for multiple comparisons. 1H, one-handers; CTR, controls; asterisks denote significance at the level of **p < 0.01.

connectivity between their cortical hand regions (r = 0.66, p = 0.004, randomisation test, middle panel of *Figure 2C*), resulting in relatively normal (symmetrical) patterns of inter-hemispheric connectivity in the individuals that adaptively use their residual arm.

## Discussion

Here we report alterations in structural and functional brain symmetry of the sensorimotor network in a large sample of individuals born without a hand compared to controls. We find reduced symmetry of white matter microstructure in the corticospinal tracts of one-handers, as well as reduced symmetry of resting-state functional connectivity patterns between the sensorimotor hand regions (i.e., reduced inter-hemispheric connectivity). Importantly, although we detected asymmetrical structural and functional patterns across the group of one-handers, we were also able to observe more symmetrical

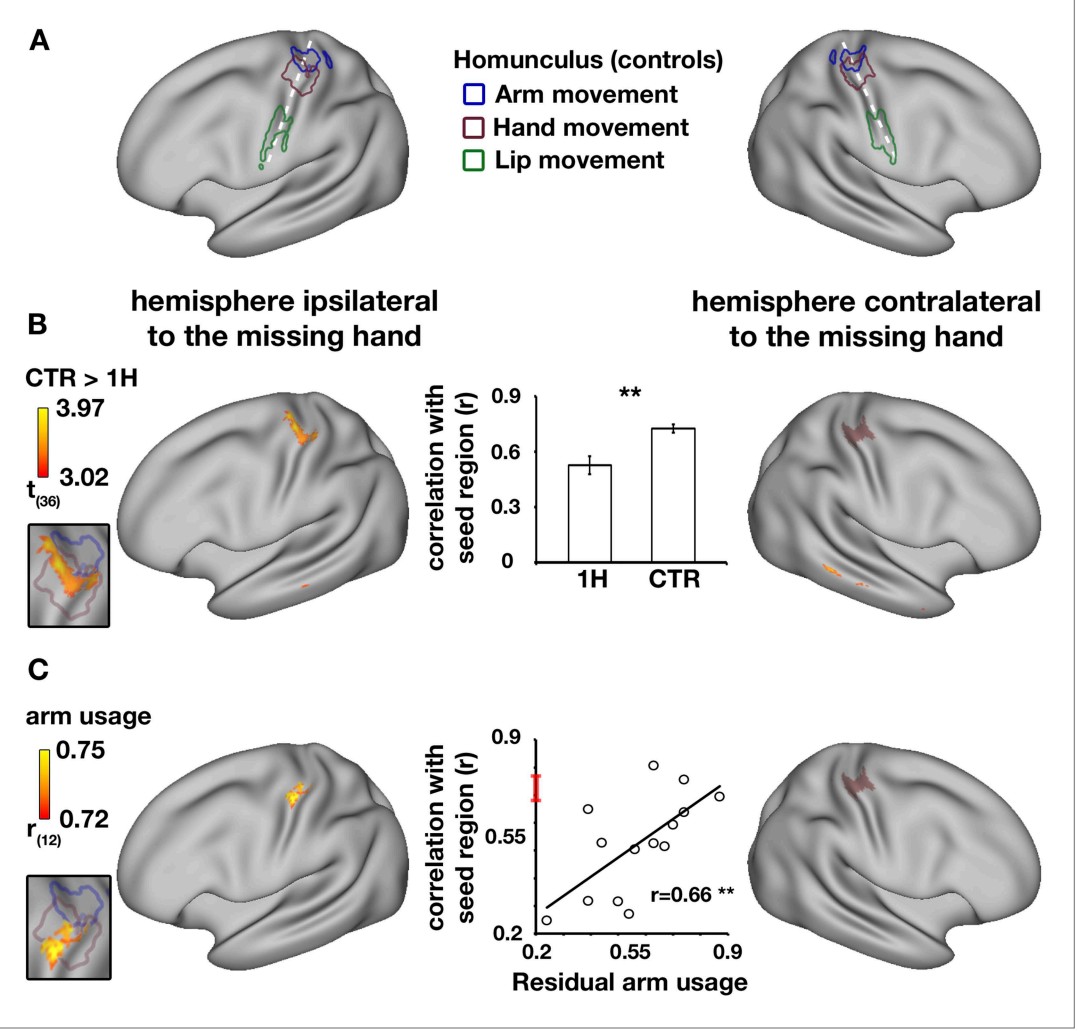

**Figure 2**. Decreased functional asymmetry in hand region functional connectivity during rest is associated with increased residual arm usage in daily tasks. (**A**) An illustration of the human sensorimotor homunculus, projected on a cortical surface map. Coloured lines show brain areas activated during execution of movements of the arms (blue), hands (purple), and lips (green) in the control group, using task-evoked fMRI scans. (**B**) Whole-brain group comparison (controls > one-handers) of resting-state functional connectivity using the missing hand's cortical territory as the seed-region. The seed-region (the mirror projection of intact/dominant hand activity, averaged across groups) is shown in purple shading. The orange-yellow clusters show areas with lower levels of connectivity to the missing hand territory in one-handers (corrected, p < 0.05). The insert on the left shows that the main resulting cluster overlaps with the task-evoked hand region, as measured in controls in (**A**), suggesting lower levels of inter-hemispheric connectivity between the cortical hand regions in one-handers. The bar plot in the middle reflects group-wise mean (±s.e.m.) connectivity levels between the seed-region and the intact hand ROI (defined based on intact/dominant hand activity, averaged across groups). (**C**) Whole-brain correlations between connectivity levels with the missing hand's seed-region (purple shading) and level of one-handers residual arm usage in daily tasks (level of bimanual usage). Significant correlations were restricted to the anterior aspect of the intact hand knob of the central sulcus, as can be seen in the insert to the left. Scatter plot in the middle reflects connectivity levels between the seed-region and the intact hand region (y axis) versus residual arm use (x axis). The red line on the y axis shows the mean ± confidence interval of controls' hand region inter-hemispheric connectivity. The significant positive correlation (p < 0.005) reflects that one-handers using their residual arm more frequently to support bimanual tasks showed higher levels of hand region functional connectivity, similar to those of control participants. This suggests that increased residual arm usage to support bimanual tasks normalises the aberrantly reduced levels of inter-hemispheric connectivity in one-handers, shown in (**B**). 1H = one-handers; CTR = controls; Asterisks denote significance at the level of **p < 0.005.

**Table 2.** Cluster statistics for presented statistical maps

| Statistical parametric map | Voxels | Max value | X | Y | Z | Region |
|---|---|---|---|---|---|---|
| Hand-evoked activation (intact hand ROI) | 305 | 9.09 (Z) | −38 | −26 | 58 | Primary sensorimotor cortex, intact/dominant hemisphere |
| Hand-evoked activation | 166 | 9.84 (Z) | 22 | −50 | −24 | Intact cerebellum, lobule VI, intact/dominant hemisphere |
| Seed-based functional connectivity, between group t-test | 239 | 4.26 (t) | −38 | −24 | 60 | Primary sensorimotor cortex, intact/dominant hemisphere |
| Seed-based functional connectivity, between group t-test | 159 | 4.22 (t) | 6 | 38 | 40 | Dorsomedial prefrontal cortex, missing/nondominant hemisphere |
| Seed-based functional connectivity, between group t-test | 72 | 3.92 (t) | 70 | −36 | −2 | Middle temporal gyrus, missing/nondominant hemisphere |
| Seed-based functional connectivity, between group t-test | 7 | 3.21 (t) | −68 | −26 | −12 | Middle temporal gyrus, intact/dominant hemisphere |
| Correlations between voxel-wise functional connectivity values and behavioural scores | 225 | 0.81 (r) | −30 | −28 | 44 | Primary sensorimotor cortex, intact/dominant hemisphere |

Columns present the cluster size (no. of voxels), the value of the maximum 'intensity' within the cluster (Max value), the location of the maximum intensity voxel given as X/Y/Z values in MNI coordinates, and the functional region. Intact/dominant hemisphere relates to the hemisphere contralateral to the intact/dominant hand in one-handers/controls, respectively. Missing/nondominant hemisphere relates to the hemisphere contralateral to the missing/nondominant hand in one-handers/controls, respectively.

functional connectivity patterns in some individuals. Specifically, those one-handers who utilising their residual arm for typically bimanual tasks showed normal (i.e., symmetrical) levels of resting-state inter-hemispheric connectivity. Due to the cross-sectional nature of this study, we are unable to determine the origin of the increased functional connectivity observed in those who used their residual arm more frequently. This observation may reflect preservation of initially intact inter-regional connections, which are otherwise lost through lack of use, or may instead reflect establishment of (otherwise missing) functional connections through usage. The fact that individuals with increased functional connectivity didn't also show greater structural symmetry may support the latter interpretation of functional reorganisation. Either way, our findings suggest that compensatory behaviour is a powerful driver in shaping brain organisation associated with sensorimotor deprivation, as reflected in resting-state functional connectivity. Our unique dataset also provides a rare opportunity to demonstrate how individualised ecological behaviour is imprinted in spontaneous brain connectivity patterns, as was previously hypothesised (**Harmelech and Malach, 2013**).

White matter in one-handers was relatively reduced along the corticospinal tract contralateral to the missing hand, compared to the homologous tract contralateral to the intact hand. The white matter measurement used here (FA) is an indirect and relatively unspecific measure of white matter microstructure (**Zatorre et al., 2012**; **Jones et al., 2013**). It reflects how directional water diffusion is within tissue, due to presence of structural barriers. The more coherent these barriers are, the higher the degree of diffusion directionality, that is, FA, will be. Many different features, including the degree of myelination, the fibre packing density, the membrane permeability, axon diameter and the tissue geometry, could therefore influence the reported structural asymmetry. Our observation is in line with numerous studies showing that the maturation and refinement of corticospinal pathways crucially depends on motor behaviour and the neural activity it elicits in early life (**Eyre, 2007**; **Martin et al., 2007**). Similarly, asymmetry of white matter microstructure in the posterior limb of the internal capsule has been previously shown to predict poor motor function of the upper limb after stroke (**Stinear et al., 2007**; **Lotze et al., 2012**). According to this account, reduced motor behaviour and subsequent neuronal activity in one-handers may result in decreased white matter microstructure in the corticospinal tract contralateral to the missing hand. However, as we couldn't identify any statistical relationship between residual arm usage and white matter microstructure along the corticospinal tract, this interpretation remains tentative.

The structural asymmetry in one-handers was accompanied by asymmetrical functional connectivity of the cortical hand regions (i.e., reduced inter-hemispheric connectivity). We previously reported a

reduction in hand-region inter-hemispheric functional connectivity in acquired amputees, that was associated with phantom pain (*Makin et al., 2013b*): individuals suffering from worse chronic phantom pain showed greater asymmetry in inter-hemispheric functional connectivity. We proposed that reductions in functional connectivity may relate to abnormal signalling of the peripheral injured nerve (*Vaso et al., 2014*), which could reduce the synchronicity of fluctuations between the bilateral hand regions. In the one-handers studied here, who rarely experience phantom sensations (*Price, 2006*), the observed reduction in inter-hemispheric connectivity may instead be related to the strong and chronic lateralisation of motor behaviour, resulting in dis-synchronisation of inputs between the two hand regions, as will be discussed below.

We recently showed that when one-handers move their residual (handless) arm, this results in activation within the cortical territory of the missing hand (*Makin et al., 2013a*). Based on our previous and current findings, we suggest that 'high' residual-arm users jointly activate the missing hand territory (through residual arm movements, consequential to adaptive plasticity) and the intact hand cortex (through intact hand movements) in everyday settings, such as while performing typically bimanual tasks. These repeated cortical activations are imprinted into resting-state connectivity patterns (*Harmelech and Malach, 2013*; *Guerra-Carrillo et al., 2014*), and are therefore reflected as symmetrical levels of inter-hemispheric connectivity in these individuals. The current study therefore highlights the pivotal role of the missing hand's cortical territory in use-dependent plasticity. We note that significant group differences in functional connectivity were not entirely exclusive to the intact hand region (e.g., prefrontal cortex, *Table 2*). However, effects in these other cortical areas were not related to compensatory behaviour (*Figure 2C*). Therefore, additional research will be needed in order to understand the functional role of brain regions beyond the primary sensorimotor cortex in one-handers.

The association we established here between behaviour and functional connectivity patterns in the primary sensorimotor cortex also contributes to interpretation of resting-state activity more generally. In contrast to lateralised activation patterns evoked by unimanual movements, resting-state patterns typically show strong bilateral coherence, resulting in highly symmetrical networks (*Biswal et al., 1995*; *Yeo et al., 2011*). Our observation that reduced bimanual movements (in the absence of one hand, and with limited compensatory capacity of the residual arm) is related to decreased inter-hemispheric coherence in one-handers, provides compelling new evidence that the symmetrical resting-state coherence typically found in the sensorimotor network is likely to be due to coordinated body movements in naturalistic daily settings. We therefore propose that spontaneous coherence seen in brain networks during rest (*Nir et al., 2006*) relates to the relatively integrated activity of network sub-regions during complex real-life behaviours.

## Materials and methods

### Participants

This study relied largely on the same population recruited for a previous study, using the same scanning procedures and exclusion criteria as described before (*Makin et al., 2013a*). Data from one control participant were discarded from the resting states fMRI analysese due to excessive head movements (>3 mm), and data from a different control participant were discarded from diffusion MRI analyses due to problems during data collection. Additionally, three one-handers and three control participants were recruited, resulting in a total of 14 one-handers (age = 36.64 ± 12.02, four with absent right hand, see *Table 1* for demographic details) and 24 controls (age = 41.12 ± 12.86, eight left hand dominant). Recruitment was carried through the Oxford Centre for Enablement and Opcare in accordance with NHS national research ethics service approval (10/H0707/29), and written informed consent was obtained.

### Limb-use strategy measurements

Use of residual arm and prosthesis in one-handers were assessed with a revised version of the Motor Activity Log (*Uswatte et al., 2006*), as described before (*Makin et al., 2013a*). In short, participants with limb absence were requested to rate how frequently they incorporate their residual arm (either directly, or using a prosthesis) in an inventory of daily activities, with varying degrees of motor control. This questionnaire, indexing bimanual usage, was previously validated using limb acceleration data, collected in ecological settings (*Makin et al., 2013a*).

## Scanning procedures

The scanning protocol consisted of an anatomical T1 scan, a resting-state scan, a task-evoked scan for body-part functional localisation and two diffusion-weighted imaging scans. All scanning procedures but the resting-state scan were described previously (*Makin et al., 2013a*). Resting state functional data, based on the blood oxygenation level-dependent (BOLD) signal, were acquired using a three Tesla Verio scanner (Siemens, Erlangen, Germany), and using a multiple gradient echo-planar T2*-weighted pulse sequence, with the parameters: TR = 2410 ms; TE = 30 ms; flip angle = 90°; imaging matrix = 64 × 64; FOV = 192 mm axial slices. 46 slices with slice thickness of 3 mm and no gap were oriented in the oblique axial plane, covering the whole cortex, with partial coverage of the cerebellum. A total of 128 volumes were collected (*Makin et al., 2013b*). During the scan, participants were asked to lie still for 5 min in a dimmed room with their eyes open, and let their mind wander. They were explicitly asked not to move any body-part.

## MRI data analyses

All imaging data were processed using FSL 5.1 (www.fmrib.ox.ac.uk/fsl) (*Jenkinson et al., 2012*). Diffusion and functional MRI data collected from one-handers with absent right limbs (n = 4) were mirror reversed across the mid–sagittal plane prior to all analyses so that the hemisphere corresponding to the missing hand was consistently aligned. Data collected for left-hand dominant controls (n = 8) was also flipped, in order to account for potential biases stemming from this procedure. Note that the proportion of flipped data was not different between experimental groups ($\chi^2_{(1)} = 0.18$, p = 0.67).

## Diffusion MRI data analyses

Diffusion data were corrected for eddy-currents and head motion using an affine registration to a non-diffusion-weighted volume. The diffusion tensor model (*Basser et al., 1994*) was fitted at every voxel to derive fractional anisotropy (FA) maps. The FA reflects how directional water diffusion is within tissue, due to presence of structural barriers. In the text we interchangeably refer to FA and 'white matter microstructure', but we stress that the latter terminology corresponds to a simplified interpretation.

The estimated FA maps were non-linearly registered to a standard MNI template. The resulting FA images were used to create a mean 'skeleton' (FA thresholds >0.2), representing the centre of all white matter tracts onto which participant-specific FA values were projected (*Smith et al., 2006*).

To compare the laterality of the corticospinal tracts between experimental groups, the group FA skeleton was masked with the left and right corticospinal tracts defined from the Johns Hopkins University white matter tractography atlas. In order to ensure that we compare the same white matter voxels across participants, mean FA values were extracted from the data of each participant under these left and right skeletonised corticospinal masks. For each participant, a white matter tract laterality index was computed as [(intact − residual)/(intact + residual)] for one-handers, and [(dominant − nondominant)/(dominant + nondominant)] for controls. Note that the proportion of participants with a left 'intact'/'dominant' hemisphere was similar across groups ($\chi^2_{(1)} = 0.18$, p = 0.67). Since laterality indices cannot be assumed to follow a normal distribution, the significance of difference between group indices was assessed using a randomisation test, under the null hypothesis of no difference between group means. To ensure that the null-distribution will not be biased by the difference if group sizes, we equated group sizes by randomly choosing a sub-sample of 14 controls for each of the below described iterations. In each iteration, participants' labels (one-handers or controls) were shuffled, creating two random experimental groups, and the difference between the averaged group indices was calculated. This procedure was repeated 10,000 times to construct the null distribution, and the position of the test's statistic in relation to this distribution was determined. To evaluate the specificity of group-differences to the corticospinal tracts, the same procedure was repeated for non-motor control tracts: the bilateral Inferior fronto-occipital fasiculi and the bilateral Inferior longitudinal fasiculi.

In order to further localise regions that drive the group white matter laterality difference, lateralities between corresponding locations along the corticospinal tracts were computed. For each participant separately and for each slice along the z-axis, a laterality index was computed as described above, using voxels of the skeletonised tracts within that slice. The resulting single-subject point–wise FA laterality indices were then compared between groups using permutation-based cluster statistics. Clusters were tested for significance using threshold-free cluster enhancement (*Smith and Nichols, 2009*) at p < 0.05,

corrected for multiple comparisons across space. For presentation purposes, FA clusters were projected onto the canonical left corticospinal tract.

## Preprocessing of functional data

Functional data were analysed using FMRIB's expert analysis tool (FEAT, version 5.98). Preprocessing of each individual task-evoked run was described previously (*Makin et al., 2013a*). For each individual resting states scan, the following pre-statistics processing was applied: motion correction using FMRIB's Linear Image Registration Tool (*Jenkinson et al., 2002*); brain-extraction using BET (*Smith, 2002*); and high pass temporal filtering of 150 s. Non-neuronal contributions to the BOLD signal were removed from the unsmoothed data of each participant by linear regression of motion parameters, as well as time-courses extracted from the ventricles and the white-matter. Data were then spatially smoothed using a Gaussian kernel of 8 mm full width at half maximum.

All functional data were aligned to structural images (within-subject) initially using linear registration, then optimised using Boundary-Based Registration (*Greve and Fischl, 2009*). Structural images were transformed to standard MNI space using non-linear registration, and the resulting warp fields were applied to the functional statistical summary images.

## Functional data analysis

All statistical analyses were conducted using FSL and in-house Matlab code (Mathworks, Natick, MA, USA). To compute statistical parametric maps, we applied a voxel-based general linear model (GLM), as implemented in FEAT. Low-level analysis is detailed below for resting-state and task-evoked datasets. Group level analysis of spatial maps was carried out using FMRIB's local analysis of mixed effects. The cross-participant GLM included planned comparisons across or between the two groups. Z (Gaussianised T/F) statistic images were thresholded as detailed below. Family-wise error corrected cluster significance threshold of $p < 0.05$ was applied to the suprathreshold clusters.

### ROI definition

ROI definition was based on a block design fMRI task, involving simple movements of the left and right hand/arm, lips and feet, resulting in six experimental conditions, as previously described (*Makin et al., 2013a*). The block design paradigm, as well as its temporal derivatives, was convolved with a gamma function (*Friston et al., 1998*) to model the activation time-course, associated with hand and arm movements (versus baseline) at the individual participant level. The contrast of main interest was the intact hand condition against baseline. The intact hand ROI was defined by averaging the low-level contrast of intact/dominant hand vs rest across the two groups (one-handers and controls). A threshold of $Z > 7.5$ was chosen, yielding a cluster centered around the hand knob of the central sulcus. As the spatial resolution and co-registration methods used here are insufficient to reliably dissociate the somatosensory and motor primary cortices, cortical regions spanning the central sulcus, and the pre- and post–central gyri were described here as 'sensorimotor'. Since the primary sensorimotor cortex shows the highest levels of functional symmetry across the two cerebral hemispheres (*Stark et al., 2008*), the ROI representing the cortical territory of the missing hand was created by mirror flipping the intact hand ROI coordinates on the x-axis.

### Homunculus construction

To visualise the typical representation of lips, hands and arms in *Figure 2*, contrasts for each task condition was defined against the feet condition in each control participant. Low- and high-level analyses were carried as described above. Each high level map was thresholded at $Z > 5.5$ and the thresholded maps were mirror flipped and summed, in order to achieve a symmetrical bilateral group map. Each lips map was divided in 2, since lip movements are bilateral. Symmetrical maps of homologous body parts were then averaged, resulting in three bilateral maps, which were thresholded at $Z > 5$. Note that due to the comparison between each body-part condition and baseline, the cluster involving the lips was merged with activation from the secondary somatosensory cortex. Resulting maps were projected on an inflated averaged brain, using the Connectome Workbench (http://humanconnectome.org/software/get-connectome-workbench.html, [*Marcus et al., 2011*]). This procedure was carried for visualisation purposes only, and was not used during data analysis.

### Resting-state statistical analyses

The cross-groups ROI of the missing/nondominant hand region (in one-handers and control participants, respectively) was used as a seed-region in a functional connectivity analysis. For each

participant separately, the averaged time-course of the voxels comprising this ROI was correlated with each of all other voxel time-courses of the brain, using MATLAB. Resulting Pearson correlation coefficients were converted to z-values using Fisher's correction to improve normality. A group disparity map was created using a two-tailed two-sampled $t$ test, using MATLAB. All maps were corrected for multiple comparisons using FDR at significance threshold of $p < 0.05$. Resulting disparity values were projected onto an inflated averaged brain, using Workbench. To assess the significance of the hand regions inter-hemispheric connectivity effect, an independent ROI analysis was conducted. To that end, the temporal correlation between the time-courses of the intact and missing hand ROIs was calculated for each participant. The resulting values were compared between groups using a randomisation test, as described before, under the null hypothesis of no group differences in inter-regional functional connectivity.

To ensure that group differences in functional connectivity were not caused by a reduced signal-to-noise ratio in the missing hand region, which may be related to low levels of neural activity in the one-handers, the variance of BOLD fluctuations within the missing hand's cortical territory was compared between groups. First, the mean BOLD signal was extracted from the seed ROI in each participant, and the variance of this mean signal was compared between participants of the two experimental groups using a randomisation test, under the null hypothesis of no group difference in the temporal variance of this region, and using the same procedure described before. To gain a finer grained estimate, the signal variance of each voxel comprising the missing hand ROI was averaged for each participant, and compared between groups using a randomisation test, as described before.

In order to assess the positive relation between hand-region inter-hemispheric connectivity levels and adaptive residual arm use, bimanual usage index scores were correlated with voxel-wise functional connectivity values, derived from the seed analysis across all one-handed participants. Resulting correlation coefficients were Fisher corrected to improve normality and a voxel-wise one-sample one-tailed $t$ test was conducted to assess the threshold of correlation significance. The thresholded correlation map was corrected for multiple comparisons using FDR at significance threshold of $p < 0.05$, and projected onto an inflated template brain, using Workbench.

To confirm the relationship between adaptive behaviour and hand region inter-hemispheric functional connectivity, we implemented an ROI approach. Mean functional connectivity values under the independently defined intact hand ROI were extracted from the single-subject maps of one-handed participants. These values were then correlated with bimanual usage index scores for these participants. To obtain a stringent estimate of significance level, the resulting correlation coefficient was subjected to a randomisation test. Specifically, the null distributions, assuming no correlation between inter-hemispheric connectivity and bimanual usage index scores, was constructed by randomly shuffling the participants' inter-hemispheric connectivity levels and correlating these with the unshuffled bimanual usage index scores. This procedure was repeated 10,000 times creating the null distribution, and the position of true correlation coefficient in relation to this distribution determined the test's p-value.

## Acknowledgements

This work was supported by the Israeli Presidential Bursary for outstanding PhD students in brain research and the Boehringer Ingelheim Fonds travel grant (AH); EU FP7 VERE grant, ISF ICORE, EU Flagship HBP, and Helen and Martin Kimmel award (RM); Wellcome Trust, the NIHR Oxford Biomedical Research Centre and EU FP7 ABC grant (PITN-GA-2008-290011) (HJB); Royal Society (Newton International Fellowship), European Commission (Marie Curie Intra- European Fellowship) and Royal Society/Wellcome Trust (Sir Henry Dale Fellowship, grant number 104128/Z/14/Z) (TRM). We thank Tim Behrens and Andy Segerdahl for helpful comments on the manuscript, Opcare for assistance with participants recruitment and our participants for their support.

## Additional information

### Funding

| Funder | Grant reference number | Author |
| --- | --- | --- |
| Israeli Presidential Bursary | | Avital Hahamy |
| Boehringer Ingelheim | | Avital Hahamy |

| Funder | Grant reference number | Author |
|---|---|---|
| European Commission | EU FP7 VERE | Rafael Malach |
| European Commission | Flagship HBP | Rafael Malach |
| Israel Science Foundation | ICORE | Rafael Malach |
| Wellcome Trust | | Heidi Johansen-Berg |
| National Institute for Health Research | Oxford Biomedical Research Centre | Heidi Johansen-Berg |
| European Commission | EU FP7 ABC grant (PITN-GA-2008-290011) | Heidi Johansen-Berg |
| Royal Society | Newton International Fellowship | Tamar R Makin |
| European Commission | (Marie Curie Intra- European Fellowship | Tamar R Makin |
| Wellcome Trust | Sir Henry Dale Fellowship | Tamar R Makin |
| Royal Society | Sir Henry Dale Fellowship | Tamar R Makin |

The funders had no role in study design, data collection and interpretation, or the decision to submit the work for publication.

## Author contributions

AH, TRM, Conception and design, Acquisition of data, Analysis and interpretation of data, Drafting or revising the article; SNS, RM, Analysis and interpretation of data, Drafting or revising the article; DHS, Conception and design, Drafting or revising the article; HJ-B, Conception and design, Analysis and interpretation of data, Drafting or revising the article

## Ethics

Human subjects: Informed consent and consent to publish was obtained in accordance with ethical standards set out by the Declaration of Helsinki (1964) and with procedures approved by the NHS (REC ref: 10/H0707/29).

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
