## [Decision Letter]

Thank you for sending your Research advance entitled “Normalisation of brain connectivity through compensatory behaviour, despite congenital hand absence” for consideration at *eLife*. Your article has been favorably evaluated by Eve Marder (Senior editor), a Reviewing editor, and 2 reviewers.

The Reviewing editor and the reviewers discussed their comments before we reached this decision, and the Reviewing editor has assembled the following comments to help you prepare a revised submission.

While the Research advance mechanism is intended as a follow-up to a previously published *eLife* paper, it is still crucial that the new paper be of the highest standards. While the reviewers found the data in the paper interesting, they found its presentation more challenging. We trust that you will be able to significantly improve the clarity of the revised version.

Major comments:

1) Reviewer #1 asks the authors to discuss the physiological findings of the fractional anisotropy (FA). In fact the reviewer is not sure the authors convey the message of FA in terms of, for example, does FA correlate with a reduced number of cortico-spinal fibers? Does it correlate with less dense white matter tracks or less dense myelin?

2) Reviewer #2 feels that the current paper was directed to experts in the field and that the paper is plagued with jargon. Also, the logic of the paper is not made explicit, which makes it difficult to read the paper.

---

## [Author Response]

*1) Reviewer #1 asks the authors to discuss the physiological findings of the fractional anisotropy (FA). In fact the reviewer is not sure the authors convey the message of FA in terms of, for example, does FA correlate with a reduced number of cortico-spinal fibers? Does it correlate with less dense white matter tracks or less dense myelin*?

We thank the reviewer for this comment. FA is a commonly used index to assess tissue microstructure. Specifically, it has been shown to be a powerful and sensitive measurement of white-matter features, as reflected in many studies involving both healthy and diseased populations. Unfortunately, being an MRI-derived index, it is an indirect measure, and as such lacks the specificity for an unambiguous biological interpretation (13; 24; 38). To answer the Reviewer’s questions, yes, FA does correlate with tract density and myelin, but also with many other microstructural features, including axon diameter, membrane permeability and the tissue geometry. FA changes can reflect changes in any of these factors individually and/or their combinations. For these reasons, in the manuscript we avoided making speculative claims about the microstructure features underlying our effect. In the revised manuscript we describe the microstructural features that might underlie the white matter measurement, and its limitations.

For example, in the Discussion section we now state:

“The FA is an indirect and relatively unspecific measure of white matter microstructure (38; 13). It reflects how directional water diffusion is within tissue, due to presence of structural barriers. The more coherent these barriers are, the higher the degree of diffusion directionality, i.e. FA, will be. Many different features, including the degree of myelination, the fibre packing density, the membrane permeability, axon diameter and the tissue geometry, could therefore influence the reported structural asymmetry. Our observation is in line with numerous studies showing that the maturation and refinement of corticospinal pathways crucially depends on motor behaviour and the neural activity it elicits in early life (5; 18). Similarly, asymmetry of white matter microstructure in the posterior limb of the internal capsule has been previously shown to predict poor motor function of the upper limb after stroke (31; 14).”

*2) Reviewer #2 feels that the current paper was directed to experts in the field and that the paper is plagued with jargon. Also, the logic of the paper is not made explicit, which makes it difficult to read the paper*.

We thank the reviewer for this important comment. We agree our paper was not pitched at a suitable level for a more general audience of non-experts. We have now rewritten substantial parts of the text, minimising use of jargon (such as “ROI”, “deprived cortex”, FA etc.) and providing clearer explanation of our rationale and results.

For example, in the Introduction section we now clearly state our predictions:

“Since it has been hypothesised that patterns of resting-state functional connectivity reflect the activity patterns evoked by every-day behaviours (10), we predicted that the way individuals behave to compensate for their disability would also be reflected in resting-state connectivity patterns. We focused on inter-hemispheric functional connectivity patterns, which are stronger between homologous regions along the sensorimotor homunculus, resulting in a symmetrical connectivity pattern. We aimed to test whether this symmetry is disrupted in one-handers who have a more asymmetrical pattern of limb use in daily life (i.e. increased usage of their intact hand versus their residual arm).”

We emphasise that the entire text has been thoroughly modified according to the reviewer’s suggestions. We feel confident these changes enhance the accessibility of our paper to non-experts.